# Carbon-Fiber-Recycling Strategies: A Secondary Waste Stream Used for PA6,6 Thermoplastic Composite Applications

**DOI:** 10.3390/ma16155436

**Published:** 2023-08-03

**Authors:** Marco Valente, Matteo Sambucci, Ilaria Rossitti, Silvia Abruzzese, Claudia Sergi, Fabrizio Sarasini, Jacopo Tirillò

**Affiliations:** 1Department of Chemical Engineering, Materials, Environment, Sapienza University of Rome, 00184 Rome, Italy; matteo.sambucci@uniroma1.it (M.S.); ilaria.rossitti@uniroma1.it (I.R.); abruzzese.1638778@studenti.uniroma1.it (S.A.); claudia.sergi@uniroma1.it (C.S.); fabrizio.sarasini@uniroma1.it (F.S.); jacopo.tirillo@uniroma1.it (J.T.); 2INSTM Reference Laboratory for Engineering of Surface Treatments, Department of Chemical Engineering, Materials, Environment, Sapienza University of Rome, 00184 Rome, Italy

**Keywords:** short-carbon-fiber-reinforced PA6,6 thermoplastic composites, recycled carbon fibers, 3D printing, recoverability, critical length

## Abstract

With a view to achieving sustainable development and a circular economy, this work focused on the possibility to valorize a secondary waste stream of recycled carbon fiber (rCF) to produce a 3D printing usable material with a PA6,6 polymer matrix. The reinforcing fibers implemented in the research are the result of a double-recovery action: starting with pyrolysis, long fibers are obtained, which are used to produce non-woven fabrics, and subsequently, fiber agglomerate wastes obtained from this last process are ground in a ball mill. The effect of different amounts of reinforcement at 5% and 10% by weight on the mechanical properties of 3D-printed thermoplastic composites was investigated. Although the recycled fraction was successfully integrated in the production of filaments for 3D printing and therefore in the production of specimens via the fused deposition modeling technique, the results showed that fibers did not improve the mechanical properties as expected, due to an unsuitable average size distribution and the presence of a predominant dusty fraction ascribed to the non-optimized ball milling process. PA6,6 + 10 wt.% rCF composites exhibited a tensile strength of 59.53 MPa and a tensile modulus of 2.24 GPa, which correspond to an improvement in mechanical behavior of 5% and 21% compared to the neat PA6,6 specimens, respectively. The printed composite specimens loaded with the lowest content of rCF provided the greatest improvement in strength (+9% over the neat sample). Next, a prediction of the “optimum” critical length of carbon fibers was proposed that could be used for future optimization of recycled fiber processing.

## 1. Introduction

In recent years, carbon fibers (CFs) have found applications in a wide variety of fields, including automotive, aircraft, electronic equipment, and sporting goods industries [1]. Their use as reinforcement in composite materials is constantly growing even in fields more oriented toward mass production by replacing traditional metallic components, providing highly specific mechanical properties that result in lower CO_2_ emissions [2]. Data analyses of this trend suggest that the world production of CFs already doubled between 2009 and 2014 from 27 to 53 ktons and reached a maximum in 2022 with 117 ktons [3].

Such a boost in the use of CFs affects the generation of wastes that derive both from the production process (process wastes representing about 30–40 wt.% of the total) and from the end of life of the products [1]. Currently, the main ways for the disposal of composite waste are incineration and landfilling, but these strategies will no longer be adequate due to environmental pollution and loss of CFs with high added value [2]. Recently, several recycling technologies have been proposed to treat composite materials with CFs and recover the CFs to reuse them in other applications for both economic and environmental reasons, aiming at the production of sustainable composites made with recycled reinforcement and a recyclable matrix [4]. The main technologies explored so far include mechanical recycling (shredding, crushing, milling), chemical recycling (solvent, catalyst, or supercritical fluids), and thermal recycling (pyrolysis, oxidation, steam thermolysis), and the objective of these techniques is to recover CFs under conditions as close to their initial state as possible to facilitate reuse in other applications [5]. Mechanical recycling is currently one of the consolidated methods and involves several steps to reduce the size of waste. First, the composites are ground to a size of about 50–100 mm, and then, further grinding is applied to obtain recycled materials with different dimensions, which can range from powder to fibrous agglomerates. The materials obtained from this type of recycling can be used as reinforcement in short-fiber composite materials, such as those used for extrusion and injection molding. Because of the resultant limited fiber aspect ratio, these materials have a low market value [6]. Due to the friction caused by materials during recycling, damage to equipment can occur, which directly increases the cost of various operations, and this decreases the economic margin of recycled materials, often making this choice less feasible. Dust produced by the recycling system is a major safety and health hazard, but it can be easily reduced with engineering controls and good ventilation [1,6]. In the case of chemical recycling, called solvolysis, the polymeric matrix is decomposed with a solution of acids, bases, and solvents whose composition must be adjusted according to the nature of the matrix. To increase the contact surface with the solution and aid in the dissolution of the matrix, solid composites are ground first. At the end of the process, the CFs are washed to remove the decomposed polymer residues and the solvent residues.

Recycled CFs obtained in this way can be longer than mechanically ground ones and have been shown to maintain their tensile strength, with only a small percentage less than virgin CFs. However, the use of hazardous and concentrated chemicals has a significant environmental impact [7]. In thermal recycling, high temperatures are used to degrade the polymeric matrix and leave the CFs as a residue, and it can be divided into different types. For thermal processes, the operating parameters must be carefully controlled to avoid the loss of valuable products or a change in the chemistry of the recovered materials. If the process temperature is too low, the fiber surface is covered by a layer of amorphous carbon following the limited degradation of the matrix, whereas if the temperature is too high, the CF surfaces oxidizes, with consequent reduction in the diameter of the fibers and mechanical properties in general [2,8,9]. Therefore, the best way is first to recover long fibers through the process of pyrolysis, which will be later used to produce semi-finished products, such as non-woven fabrics, to be impregnated in a subsequent step. This step, in turn, generates waste, namely short fibers. The latter, through further processing, could be used in fused deposition modeling (FDM) technology. The mechanical properties of composite materials depend on many factors, and the length of the fibers must certainly be mentioned among the main ones [10]. Normally, short-CF-reinforced composite materials are prepared using extrusion compounding and injection molding techniques, and in addition to the initial length of the fibers, it should be remembered that during the production of these composites, fiber breakage occurs due to fiber–fiber interaction [11].

The manufacturing of filaments for 3D printing also falls within the scope of extrusion, which is the focus of this study. Three-dimensional printing has already been successfully applied in the manufacture of polymeric components ranging from prototypes to final products, but the main problem is that the resulting parts have inferior mechanical performance compared to parts fabricated using conventional techniques, such as injection molding. To solve this problem, it has been seen that the addition of fibers to the polymer matrix forming a composite produces a significant improvement in the resistance of the molded parts [12]. At present, thermoplastic polymers are the ones mostly used in these processes, including acrylonitrile butadiene styrene (ABS), polylactic acid (PLA), polycarbonate (PC), and polyamide (PA).

In recent years, many authors have studied the addition of short fibers to a thermoplastic polymer to create filaments of the composite material used as raw material [13]. All these studies have analyzed the effect of fiber content, fiber orientation, and fiber length [14] on the processability and performance of the manufactured components. In addition, some studies have reported comparisons between the properties of 3D-printed composites and those manufactured with traditional molding techniques [12,15]. Among them, the work of mechanical and morphological characterization and of PLA reinforced with 15 wt.% of short CFs carried out by Ferreira et al. [14], the study of the effects of process parameters on the tensile properties of parts fabricated with ABS and CFs by Ning et al. [16], and the investigation of engineering applications of PEEK composites and short CFs by Wang et al. [17] can be worth mentioning. In their work, Ferreira et al., however, show problems in the adhesion between PLA and CFs, and embrittlement of the reinforced material due to the addition of short CFs is reported [14]. In the study by Ning et al., the effects of process parameters on the tensile properties of the composites made with FDM are described, which have much in common with the composites made in the context of this study with the additive manufacturing technique [16]. Wang et al. in their research instead deal with the production of composite materials with PEEK and CFs with a variable fiber content between 5 and 15% by weight successfully produced with extrusion [17]. In addition, the work by Giani et al. [18], in which the applicability of recycled carbon fibers to produce CF-reinforced PLA composites is demonstrated, is also worth mentioning. In this case, the comparison between the neat and reinforced samples showed about a doubling of the elastic modulus and maximum stress. All these results are particularly interesting, given the ease of manufacturing filaments for FDM and 3D printing, and confirm the possibility of applying CFs in these areas, also by exploiting different matrices as environmental and economically sustainable alternatives.

The carbon fibers used in this study fall into the category of waste generated directly by the production process that uses waste material, and the main aim lies in the identification of a potential application for this by-product considered a processing waste up to now. Given this fact and to fall within the scope of eco-sustainable and circular composite materials, the objective of this work is to study the characteristics and mechanical properties of composite materials obtained by combining a PA6,6 thermoplastic matrix with doubly recycled carbon fiber reinforcement by exploiting the additive manufacturing process. The result of this work contributes some knowledge about converting by-products to added-value 3D-printable composites with improved performance. To the best of the authors’ knowledge, investigations on the influence of fiber size on CF-reinforced filaments for 3D printing are still insufficient. Few studies have dealt with the effect of fiber length downsizing due to recycling processing on the mechanical behavior of filaments and 3D-printed parts. This article addresses the issues related to the size of the fibers used and the presence of a predominant dusty fraction resulting from the milling treatment of pyrolyzed carbon fibers to obtain fillers suitable for the 3D-printable compounding. Filaments made of neat PA6,6, as a reference material, and filaments loaded with 5 wt.% and 10 wt.% of rCFs were manufactured. All filaments were studied in terms of mechanical properties, microstructure, fiber–polymer composition, and fiber size distribution. Next, 3D-printed specimens were produced and characterized via mechanical and morphological analysis. The findings of the research aim to establish a link between recycler and FDM user by providing information to optimize the processing of rCFs for maximizing the mechanical performance of the final product.

## 2. Materials and Methods

### 2.1. Materials

The materials used in this study were polyamide BASF Ultramid^®^ 1000-11 NF2001 PA6,6 (BASF, Ludwigshafen, Germany) used as a matrix and carbon microfibers supplied by Carbon Task Srl (Biella, Italy) used as reinforcement for the composite. The characteristics of the raw polymer material are listed in Table 1.

The carbon microfibers (7 μm diameter) were supplied by Carbon Task Srl (Biella, Italy), a company that deals with the weaving of dry CFs from production waste and that, through its own production line designed over the past four years, has created a process that allows the production of non-woven fabrics of various weights.

These microfibers are the result of a double-recovery action, starting from long-fiber recycled carbon waste deriving from the pyrolysis of fiber-reinforced composites, subsequently sent to the production of non-woven fabric. The conversion of the fibers into woven-non-woven tissue carbon, including carding, fiber pile laying, and woven bonding/fixation, produces processing scrap in the form of agglomerates of microfibers (average length 550 µm), which are separated using cyclone filters. The agglomerates are collected and then ground using ball milling, obtaining the rCF filler used as reinforcement for the PA6,6 matrix in this work. Figure 1 shows a schematization of the double-recovery action that leads to the formation of carbon microfibers.

The physical properties and chemical composition of these microfibers were obtained experimentally. Their density was determined using a helium pycnometer (ASTM UOP851) and was found to be 1.917 g/cm^3^. Subsequently, scanning electron microscopy (MIRA 3 FEG-SEM, TESCAN, Brno, Czech Republic) and energy-dispersive spectroscopy (Octane Elect EDS system, Edax, Mahwah, NJ, USA) analyses of the microfibers were carried out to study their morphology and elemental composition. As can be seen in Figure 2, the sample was characterized by numerous fragments of CFs with a varied granulometry and a predominant fraction of pulverulent material, due to the ball milling process to which the raw materials were subjected.

Preliminary measurements were carried out using the SEM analysis program to evaluate the average size of the fiber fragments, which reported an average length of 30 ± 12 µm.

Regarding the EDS analysis, traces of aluminum and oxygen (Figure 3) were identified.

When analyzing the relative amounts of aluminum and oxygen, it was noted that they were present in a ratio of about 2:3 (Table 2), and therefore, it could be assumed that they were traces of alumina (Al_2_O_3_) probably deriving from the mill balls, which are made of alumina.

#### 2.1.1. Filament Extrusion for 3D Printing

Three types of filaments with different amounts of fibers were produced:Neat PA6,6 filamentFilament of PA6,6 + 5% rCF, i.e., with the addition of 5 wt.% of recycled CFsFilament of PA6,6 + 10% rCF, i.e., with the addition of 10 wt.% of recycled CFs

A die system was connected to a Thermo Scientific Process 11 twin-screw extruder (Thermo Fisher Scientific, Waltham, MA, USA), in which the material leaving the extruder is passed through a series of rotating cylinders that allow the filament to be given a constant pre-established cross section to reach the final phase of winding on the reel.

Different extrusion temperature profiles were selected for neat and composite filaments, given the presence of carbon microfibers that could lead to alterations in the rheology and thermal properties of the mixture. Table 2 reports the temperature programs inside the extruder (from feed to die) used in this work.

In all cases, the diameter of the filaments was periodically checked during the winding phase to be sure that it fell within the dimensional range allowed for feeding to the 3D printer, i.e., a diameter between 1.60 and 1.80 mm, with an optimum of 1.75 mm [19,20]. The filaments’ diameters were measured using a digital caliper (sensitivity of 0.05 mm). Figure 4 illustrates the PA6,6 filaments obtained from the extrusion process.

#### 2.1.2. 3D Printing

After the in-depth study of the properties and characteristics of the 3D printing filaments, the 3D printing of dumbbell-shaped samples with the three formulations chosen was carried out in order to perform tensile tests. The machine used was a PRUSA i3 3D printer (Prusa, Prague, Czech Republic) (Figure 5a). The standard dumbbell-shaped specimens were 7 mm × 3 mm, with a useful length of 45 mm, with curved fittings and no perimeter (Figure 5b). As far as the operating temperatures are concerned, a printing temperature of 265 °C and a bed temperature of 90 °C were used for the neat PA6,6 and 5 wt.% rCF specimens, while for the 10 wt.% rCF specimens, a printing temperature of 270 °C and a bed temperature of 110 °C were used. A printing speed of 40 mm/s, 100% infill percentage, an aligned rectilinear pattern, and a layer height of 0.2 mm were set for all three families. Before printing, filament spools were dehumidified in a drying oven for 24 h at 80 °C.

### 2.2. Methods

#### 2.2.1. SEM Analysis and Sample Preparation (Carbon Sputtering Technique and Cryogenic Fracture)

As the first step, fracture surfaces of the three types of filaments used were analyzed, after brittle fracture in liquid nitrogen, using SEM, namely the neat filament of PA6,6 only and the pair of composite filaments with 5 and 10 wt.% reinforcement in recycled CFs. Prior to the SEM analysis, the specimens were sputter-coated with carbon to make the material conductive for the analysis. This pre-treatment was performed using an EM SCD005 vacuum sputter coater (Leica, Wetzlar, Germany).

#### 2.2.2. Density Measurements

Regarding the characterization methods, the effective percentage of fiber by volume in the manufactured composite filaments was first evaluated by means of density measurements, assuming that the dispersion of fibers was homogeneous within the composite material. The density of rCFs was determined using an AccuPyc II 1340 helium pycnometer (MICROMERITICS, Norcross, GA, USA), where the volume of a sample is estimated from a pressure change gradient upon expanding of a fixed amount of helium to a reference chamber of a known volume based on ideal gas law (ASTM UOP851). The system is accurate to within 0.03% of reading values [21]. The density of the composite filaments was measured in accordance with the Archimedean principle (buoyancy method in water) [22] using a commercial density determination kit of a 0.1 mg resolution analytical balance Mettler Toledo ME54 (Mettler Toledo, Worthington, OH, USA). The mass of the specimens was weighed in air and distilled water, and density (ρ) was computed according to Equation (1):(1)$$\rho =\frac{M_{a}\times \rho_{w}}{M_{a}-M_{w}}$$
where ρ is the density of the sample, M_a_ is the mass of the sample in air, M_w_ is the mass of the sample in water, and ρ_w_ is the density of water at the measured temperature. For each composite, 20 filament specimens (~3 cm length) were tested. The results obtained were then compared with those obtained with chemical digestion, carried out following the ASTM D3171 standard. The procedure was carried out using 95–97% sulfuric acid (Honeywell Fluka, Charlotte, NC, USA) and 30% hydrogen peroxide (J. T. Baker, Waltham, MA, USA).

#### 2.2.3. Study of the Dimensional Distribution of the Fibers

To improve the filament production process with a view to future optimization, the effective dimensional distribution of the carbon microfibers was studied.

The dimensions of a fiber sample not yet subjected to any production process and fiber samples obtained following chemical dissolution of the matrix with formic acid on small portions of filaments (both at 5 and 10 wt.% of CFs) were analyzed. In both cases, portions of microfibers were isolated and placed on slides and the dimensional study was carried out using a Leica DMI5000 M optical microscope (Leica, Wetzlar, Germany) and Image J imaging software (version 2.3). A total of 20 images for each sample was captured, and a total of 90 measurements were made.

#### 2.2.4. Tensile Testing of the Filaments

The filaments were tensile-tested with a Zwick/Roell universal testing machine (Z010, load cell: 10 kN, preload: 10 Mpa, Ulm, Germany) with a crosshead speed of 20 mm/min and a gauge length of 30 mm. Before carrying out the test, filament specimens (length of 150 mm) were conditioned in an oven at 80 °C for at least 24 h to avoid problems related to the moisture absorption tendency of the PA6,6 matrix. For each formulation (neat PA6,6, PA6,6 + 5% rCF, and PA6,6 + 10% rCF), 5 filament samples were tested.

The success of these tests is appreciable since no filament, during the execution of the tests, experienced failure or damage at the clamps but rather underwent necking, as in the best of desirable cases (Figure 6).

#### 2.2.5. Study of Fiber Homogeneity within the Filaments

To verify the uniformity of fiber distribution throughout the entire filament, the density of each filament type was measured with a scale that uses the principle of hydrostatic buoyancy. Samples about 3 cm in length were prepared by cutting the filament in different areas. Table 3 shows the average value of the calculated densities for each family (20 test specimens for each material) and their standard deviations.

The small values of the standard deviation are indicative of the homogeneity of fiber content along the filaments.

#### 2.2.6. Tensile Tests of the 3D-Printed Samples

Four specimens from each family were subjected to tensile tests with the equipment already used for testing single filaments (Zwick/Roell Z010 universal testing machine). The test was carried out in accordance with the ISO 527-4 standard. The test parameters were a crosshead speed of 5 mm/min, a gauge length of 20 mm, and a preload of 1 MPa. Strain was measured with a contacting extensometer. As for the printed filaments, the tensile test was preceded by an oven-drying treatment of the specimens at 80 °C for 24 h. For each material, the average values of tensile strength and Young’s modulus were obtained by testing on 4 specimens.

#### 2.2.7. Microstructural Analysis

After mechanical characterization, the microstructural analysis of the cross sections of some 3D-printed samples from each family was performed.

As already described for filaments, in this case also, liquid nitrogen was used to image with SEM the fracture surfaces of the specimens.

## 3. Results

### 3.1. Preliminary Analysis of PA6,6 Neat and Composite Filaments for 3D Printing

Before studying the mechanical properties of the 3D-printed specimens, the morphology and properties of the extruded filaments were investigated.

SEM analysis was carried out to study the surface of the filaments and check for any defects; tensile tests were carried out, and the actual dimensions of the carbon microfibers were studied before and after extrusion.

#### 3.1.1. SEM Analysis of Filament Fracture Surfaces

From the observation of the cross section of the filament of neat PA6,6 (Figure 7), no particular features emerged. The surface was typical of a brittle fracture mechanism (caused by cryogenic fracture), and defects, such as porosity or cavities, could not be detected. The filament showed smooth and round edges, with no air bubbles, indicating the successful removal of humidity during the oven-drying pre-treatment. The diameter was priorly measured with the digital caliper (20 measurements made along the filament spool) and was found to be 1.72 ± 0.10 mm.

Also, the filaments of PA6,6 with 5 wt.% and 10 wt.% of recycled carbon fibers displayed proper circularity and were free of a porous surface (Figure 8a,b). In these cases, at higher magnifications, carbon fibers were found on the surface, which were perfectly incorporated within the polymeric matrix, as visible in Figure 8c. This result is indicative of a proper selection of extrusion parameters for manufacturing filaments implementing single-step extrusion processing. Indeed, one of the crucial challenges in the fabrication of fiber-reinforced polymeric filaments for 3D printing is to ensure a homogeneous microstructure free of scattered porosity and fiber agglomerates. Generally, a double-extrusion cycle is required to achieve a uniform microstructure and adequate mixing/distribution of fibers within the matrix [23]. However, this additional stage can be time-consuming, inducing further degradation mechanisms (thermal and mechanical degradation) in the constituent materials [24].

#### 3.1.2. Evaluation of the Effective Percentage of Reinforcement

The extrusion of polymer composites faces two crucial aspects: (1) obtaining a homogeneous polymer–fiber compound, with the aim of fabricating filaments with reinforcements well dispersed within the matrix, and (2) evaluating the real percentage of fibers integrating into the final composite, considering possible losses of materials that occur during processing into the extrusion line [25]. In order to obtain a material as homogeneous as possible within the filament, powdered PA6,6 was chosen as the raw material to make easier the mixing of the two phases. If pelleted PA6,6 had been used, it would have been difficult to homogenize the microfibers in the extruder.

To estimate the actual amount of CFs present in the manufactured 3D printing filaments, two parallel studies were carried out: the first using the densities of the raw materials involved and the composites produced to estimate the effective percentage of reinforcement by volume and the second, following what is described by the ASTM D3171 standard, carrying out an acid attack on the filament sections and measuring the quantity of residual fibers after filtration and drying processes.

For the first method, the densities of both raw materials used and the compounds themselves were taken into consideration, calculated using a balance that exploits the principle of hydrostatic thrust, while for the case of recycled carbon microfibers, the value was calculated using a helium pycnometer.

The density values and the percentage of fibers obtained at a theoretical level using the rule of mixtures are shown in Table 4.

The study of the actual percentage of carbon fibers present inside the composite filaments was performed using chemical digestion, as mentioned, following what is described by the ASTM D3171 standard, whose application is also illustrated in the article by Bowman et al. [26].

At the end of the various chemical etching phases, by applying the formulas given in the standard itself, it is possible to obtain the reinforcement content as a percentage by weight and by volume, the matrix content by weight and by volume, and the void volume fraction. In this specific case, the aim was to calculate the value of the reinforcement content as a volume percentage to compare the results with those obtained with the previously described density method. The values obtained are shown in Table 5.

The estimate of the volumetric percentage of CFs using the rule of mixtures (Table 4) differed from the values obtained using chemical digestion (Table 5). Primarily, the divergence can be attributed to the density measurements on the composite, which include the presence of voids in the samples. In contrast, chemical digestion treatment makes the measurement independent of the material’s porosity, providing an almost real measure of the amount of reinforcement incorporated into the composites. Converting the designed weight fractions of 5 wt.% and 10 wt.% to the corresponding volumetric percentages resulted in volume contents of about 3.03% *v/v* and 6.20% *v/v*, respectively. Next, net of porosity, the filaments were loaded with an amount of reinforcement close to the target values. This finding indicates that the manufacturing process implemented in this study allowed proper control of the rCF content in the matrix.

#### 3.1.3. Dimensional Study of Carbon Microfibers

With the aim to optimize the filament production process, the actual dimensional distribution of the carbon microfibers involved in this study was investigated.

The size of a sample of fibers deriving from the ball milling process and the samples obtained following matrix dissolution were analyzed and compared. Figure 9 shows the acquired micrograph of CFs for the assessment of fiber size distribution.

Figure 10 plots the fiber size distribution for as-received rCFs and fibers from the two composite samples. As expected, the average fiber length reduced following extrusion, due to the breakage that rCFs experienced during processing. Moreover, the greatest dimensional reduction occurred in the sample loaded with a higher rCF volume fraction. The mean fiber length was reduced by 23% and 37% in PA6,6 composites incorporating 5 wt.% and 10 wt.% of CFs, respectively. This trend was predictable and is justifiable due to the brittle behavior of carbon fibers. During the extrusion process, the melting of the polymer in twin-screw extrusion requires a large energy input, generates high local stresses and strains, which are detrimental to the fiber’s dimensions [27]. In addition, a higher fiber content leads to a higher damage to the fiber size. The increased deterioration of fiber length for a higher rCF volume fraction is mainly attributed to the higher fiber–fiber interaction [28]. As the percentage of fibers added inside the dosing hopper increases, the stresses due to the interaction with the neighboring fibers also increases, enhancing breakage phenomena.

#### 3.1.4. Tensile Tests of 3D Printing Filaments

The tensile test results are shown in Figure 11. As expected, the addition of rCFs resulted in an increase in Young’s modulus and tensile strength values. Compared to the neat PA6,6 filament, the best performance was obtained for the composite specimens incorporating the highest fiber content (PA6,6 + 10 wt.% rCF), resulting in an increase up to 25% and 11% in the elastic modulus and strength, respectively. It is widely known that implementing reinforcing materials (such as CFs) for designing 3D-printable polymer composites overcomes the strength limitations of FDM-fabricated pure thermoplastic parts, while improving the functionality and applicability of the material [16]. The filaments produced in this work show promising mechanical performance when compared with similar commercial products obtained from virgin microfibers. For instance, the study conducted by Al-Mazrouei et al. [29] involved the use of a commercial 3D-printable nylon/CF filament (20 wt.% CF) having a declared tensile strength and Young’s modulus of 66.3 MPa and 2.76 GPa, respectively.

### 3.2. Characterization of Tensile Properties of Neat PA6,6 and Resulting Composites

#### 3.2.1. Tensile Tests of 3D-Printed Specimens

Following the in-depth study of the properties and characteristics of the filaments for 3D printing produced by extrusion, the manufacturing and mechanical characterization of the printed dumbbell-shaped samples was performed. Figure 12 shows the average values obtained during the tensile tests.

Tensile tests revealed that the role of microfibers in strengthening the PA6,6 matrix is similar to that detected during the filament characterization but clearly involves lower strength and stiffness values. Compared to bulk specimens, the layering effect in 3D-printed parts introduces anisotropy and reduces the capacity to resist tensile and shear load [30]. In addition, the improvement in strength was not regular with increasing rCF volume fraction. The sample filled with 10 wt.% of rCF, although with the maximum increase in the elastic modulus compared to the neat matrix (+21%), provided lower mechanical strength performance than the PA6,6 + 5 wt.% rCF sample. Indeed, the lowest dosage of rCFs induced the greatest improvement in tensile strength (+9% over the neat sample). The reason behind this behavior must be attributed to the presence of a large dusty fraction inside the microfiber feedstock used in this research and to the significant fiber breakage that occurs during compounding by extrusion and printing. This is in accordance with what was found in rCF size distribution analysis (Figure 10). It is well documented that for enhanced mechanical performance of fiber-reinforced composites, in general, a higher residual-length-to-diameter (l/d) ratio is better. As claimed by Lewicki et al. [31], the use of high-l/d-ratio (l/d > 50) CFs (~350 µm length) is necessary to achieve high-performance 3D-printable composites, while retaining a high degree of feature resolution. A proper choice of CF size can help increase the strength and stiffness of the thermoplastic polymer matrix significantly. Zhang et al. [32] found an increment of more than 40% and 140% in the tensile strength and elastic modulus of the acrylonitrile butadiene styrene (ABS) matrix, respectively, by integrating 15 wt.% of short CFs. Liao et al. [33] investigated the effect of CFs on PA12′s performance, detecting an increase in the tensile strength and stiffness of up to 102% and 266%, respectively, following the addition of 10 wt.% of carbon fibers content reinforcement. Dul et al. [34] studied the effect of short CFs (15 wt.%) on the mechanical properties of a 3D-printed PA composite, revealing a maximum tensile strength of 96 MPa and a modulus of 7.9 GPa, with an increment of +34 and +147%, respectively, when compared to the neat PA sample. Therefore, the poor reinforcement effect was primarily attributable to the inadequate l/d ratio (l/d~4) of microfibers used in this work, which tends to undergo further decrease with extrusion processing. This evidence should be crucially taken into consideration for future process optimization.

By displaying the results of the tensile tests, it is worth noting that the scattering of the results in the composite samples is significantly lower than that of the plain matrix. A possible explanation can be traced back to the influence of the thermal conductivity of the carbon microfiller. A conductive filler would promote more favorable diffusion and adhesion conditions between filaments during printing, reducing the effect of interlayer defects on the mechanical response of the material [35]. In this context, SEM analysis will clarify the influence of rCFs on the microstructure of printed samples.

In any case, the excellent behavior of all the specimens during the execution of the tests should be underlined, which, as shown in the Figure 13, underwent elongation and necking mechanisms typical of a mono-material system. The deformation and failure mode of the dumbbell-shaped specimens indicated an adequate synergy between the constituents, as well as highlighting a minimal influence of the filaments debonding on the mechanical performance of the printed material.

The stress–strain curves (Figure 14) of the neat PA6,6 sample and the PA6,6-CF composites clearly elucidate that the addition of carbon microfibers preserved the ductile behavior of the neat matrix. The stiffening induced by the rCF addition slightly reduced the elongation at break of the composites, ranging between 150% (10 wt.%. rCF) and 170% (5 wt.% rCF).

#### 3.2.2. Microstructural Analysis

Following tensile tests, microstructural analysis of the cross sections of the 3D-printed samples from each family was carried out. Regarding the surface of the neat PA6,6 specimen (Figure 15), it can be said that it was almost homogeneous; the printing filaments showed good mutual adhesion, to the point of being almost completely fused to each other, as can be seen from the presence of small inter-filament cavities.

Even the surface of the PA6,6 specimens with 5% and 10% by weight of rCFs was quite homogeneous (Figure 16a,b), with the printing filaments showing good mutual adhesion. At the same magnification as in the previous case, the triangular-shaped voids formed between the beads in the printed parts were barely distinguishable.

From a first visual analysis, it was clear that the inter-filament void fraction of the specimens loaded with carbon microfibers was lower and less widespread than that of the pure material. By processing the SEM micrographs with ImageJ software, the percentage of voids in the matrices was estimated. The void fraction in the neat PA6,6 sample was around 2.20%. In the composite specimens, the porosity rate decreased to 0.35% and 0.30% for 5 wt.% rCF and 10 wt.% rCF, respectively. A similar effect was observed by Tekinalp et al. [36], who investigated the void formation in 3D-printed polymer composites incorporating short CFs. This result would consolidate the hypothesis, discussed in Section 3.2.1, concerning the improvement effect that conductive fibers would confer on the print quality and inter-layer densification of the material. While a potential microstructural quality improvement was observed, the CF-reinforced composite samples showed a limited increment in mechanical performance. These aspects provide a clear indication that fiber length optimization is the driving factor to achieve effective enhancement in strength behavior. Figure 17a shows the fractography of the PA6,6 + 5 wt.% rCF sample generated from the tensile tests. The fracture surface appeared completely free of any type of defect associated with the layer-by-layer production of 3D-printed parts. Figure 17b displays the fiber distribution in the matrix. In addition to the l/d ratio of the fiber, the fiber–matrix interface is one of the most influencing parameters for the final properties of the composite material, which are strongly interlinked to the ability to transfer across the interfacial zone [37]. The surface of rCFs looked smooth, and the matrix did not completely wrap around the fiber, which points to a low/moderate adhesion between polymer and carbon inclusions. Moreover, some holes were observed on the fracture surface, indicating that the fiber pull-out is the predominant mechanism of fracture rather than the inter-filament debonding (common occurrence in 3D-printed specimens). Fiber desizing due to pyrolysis treatment can affect the interfacial interaction experienced by the fibers with the polyamide matrix. However, to date, the mechanisms of adhesion between thermoplastics and reinforcing fibers are not fully understood. Some studies state that the thermal desizing process may facilitate several favorable fiber–matrix interactions (mechanical interlocking, non-covalent interactions, wetting) [38]. This is because the commercially available sizing agents for CFs are primarily tailored for thermosetting resins. Therefore, when applied to thermoplastic polymers, the sizing mechanisms cause adverse effects on the fiber–matrix interface properties. Conversely, other studies [39] have demonstrated that sizing removal is an ineffective method to enhance the CF–matrix interfacial bond (specifically for polyamide matrices). The non-optimal adhesion of the fibers to the matrix would suggest that the desizing induced by thermal recycling has no significant contribution to the interface properties. More detailed investigations are required to quantify the effect of CF–matrix compatibility on the final performance of the composite over the problem related to the l/d ratio of the reinforcement.

## 4. Optimization Hypothesis

In the light of what has been investigated, it can be said that the reinforcing action of rCFs was not significant due to the implementation of a filler with an unsuitable l/d ratio. For instance, the values of the Young’s modulus and the tensile strength in our case for the composite loaded with 10 wt.% of rCFs were, respectively, 2.24 GPa and 59.53 MPa on average, which correspond to an improvement in mechanical behavior of 21% and 5%, respectively, compared to the neat PA6,6 specimens. These values are markedly lower than results available in the literature (reviewed in Section 3.2.1) implementing CFs in 3D-printed composites.

The main cause that led to such limited strength performance is certainly the unsuitable average size of the CFs used. As already mentioned, indeed, in the reinforcement used, there is a preponderant dusty fraction, and the longer fibers are then further crushed during the various phases of extrusion of the filament and subsequent printing. As mentioned, the fibers used as reinforcement are obtained by grinding, in a ball mill, inextricable agglomerates of recycled fibers. A possibility of optimization is therefore to act on the grinding phase, studying in detail an operation that allows grinding of the agglomerates, untangling the fibers, but obtaining fibers of adequate length to improve the mechanical properties of rCF-reinforced composites.

In this regard, it is possible to obtain an ideal length measurement through studies on the mechanical behavior of short-fiber composites. First, it is necessary to understand the effect of discontinuous fibers in a polymeric matrix by studying the mechanism of reinforcement of the fibers. The fibers exert their effect by limiting the deformation of the matrix, and in this way, the applied external load is transferred to the fibers by shear at the fiber–matrix interface. In short fibers, the tensile stress increases from zero at the ends up to a value σ_max_, which would occur if the fibers were continuous. σ_max_ can be determined from Equation (2):(2)$$\frac{\sigma_{max}}{E_{f}}=\frac{\sigma_{c}}{E_{l}}$$
where σ_c_ is the stress applied to the composite and E_l_ can be determined via the rule of mixtures.

Therefore, there is a minimum fiber length that will allow the fiber to reach its full loading potential. The minimum fiber length at which maximum fiber stress can be achieved is called the load transfer length (l_t_), and the value can be determined from a force balance (Equation (3)):(3)$$l_{t}=\frac{\sigma_{max}\times d}{2\times \tau }$$
where *τ* is the shear strength of the fiber–matrix interface and *d* is the diameter of the fiber [40]. Note that *l_t_* is also a function of the stress applied to the composite.

Let us consider the case of the composite material with 10 wt.% of rCFs. To carry out the calculation and estimate an attempt value for l_t_ for our specific case, we need to extrapolate the values of σ_max_, τ, and Young’s modulus of the CFs only (E_f_).

Regarding σ_max_, as a first approximation, it can be assumed that the specimen breaks during the tensile test for pull-out, as observed from the SEM analysis. Regarding the τ value, reference was made to the experimental value obtained for a PA matrix composite reinforced with desized CFs by Kim et al. [39]. The value of τ was set equal to 24 MPa. Finally, the value of Young’s modulus for carbon fibers alone was considered equal to 230 GPa [41].

Applying the rule of mixtures (Equation (4)):(4)$$E_{l}=E_{f}V_{f}+E_{m}V_{m}\approx \;17.4\;\mathrm{GPa}$$
with E_f_ and E_m_ being the modulus of the fibers and the matrix, respectively, and V_f_ and V_m_ the volume fraction of the fibers and the matrix, respectively.

Using Equation (2):$$\sigma_{max}=E_{f}\left(\frac{\sigma_{c}}{E_{l}}\right)\;\approx \;790\;\mathrm{MPa}$$

Using Equation (3):$$l_{t}=\frac{\sigma_{max}\times d}{2\times \tau }=0.115\;\mathrm{mm}=115\;\mathrm{µm}\;$$

The fiber length estimated in the calculation widely exceeds the average size of carbon fillers integrated in the composite with 10 wt% of rCFs (~19 µm), supporting the poor performance improvement of the composite.

However, it should be noted that the ideal length of 115 µm must be the average length of the fibers already incorporated into the composite and not after the ball mill processing. It was verified that between the milled rCFs and the effective production of the composite material, the fibers undergo breaking, which leads to a maximum loss of about 37% in length (for PA6,6 + 10 wt.% rCF). To compensate for the damaging effect of extrusion, preserving an adequate l/d ratio for obtaining effectiveness in mechanical performance and ensuring adequate printability of the material, an average dimension ranging between 200 and 300 µm can be proposed. These values clearly consider the average size of the input waste carbon agglomerates, which are subjected to ball milling to obtain the filler implemented in this research.

## 5. Conclusions

The purpose of the work was to explore the possibility of implementing recycled microfibers, deriving from a secondary shredding of recycled carbon fibers, as reinforcement in the production of thermoplastic matrix composites optimized for 3D printing technology. The main results of the study are as follows:The extrusion parameters were successfully optimized to obtain composites filaments suitable for 3D printing processing.The mechanical characterization of the printed filaments revealed that rCFs increase the Young’s modulus and tensile strength of the composites by up to 25% and 11% (10 wt.% of rCFs), in comparison to the performance of the neat sample, respectively.Tensile tests of printed specimens highlighted a similar increment in strength performance (+16% in Young’s modulus and +9% in tensile strength for 5 wt.% rCFs and + 21% in Young’s modulus and +5% in tensile strength for 10 wt.% rCFs). SEM analysis showed microstructures not affected by the common defects induced by the layer-by-layer deposition of additive fabrication. This demonstrates the achievement of well-selected printing parameters for the processing of the composites developed in this work.The average length of the microfibers used in this research was estimated at 30 μm, too short for consistently improving strength. In addition, a complication factor for this case is the gradual reduction in size that the fibers undergo following extrusion. The dimensional optimization of the output rCF fraction from the ball milling process is undoubtedly a challenge to be faced in order to maximize the mechanical performance of composites.

The field of additive manufacturing is currently in continuous expansion, and this, combined with the increasing interest in the field of eco-sustainable and circular composite materials, places this research in a strategic position particularly suitable for future studies and improvements. In addition to defining a more suitable processing for rCFs to ensure greater improvements in terms of strength performance, future studies will be based on the thermal characterization of the developed material. Measuring and analyzing thermal properties are crucial in the field of 3D printing, both in terms of optimization of the printing parameters and in terms of the end use of the product. With a comprehensive know-how of optimized material characteristics and manufacturing parameters, composite users can achieve technical findings regarding the potential use of rCFs in additive fabrication. The cheaper price and lower carbon footprint of rCFs compared to virgin reinforcement are favorable for the eco-design of smart and high-performance composite parts for different industries, including robotics and automotive.

## Figures and Tables

**Figure 1 materials-16-05436-f001:**
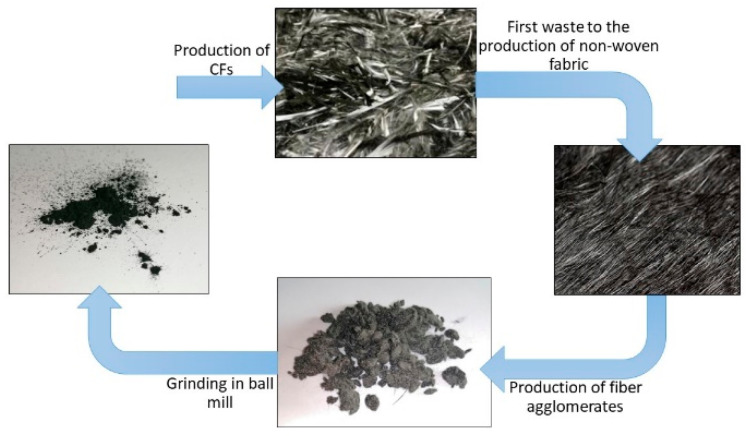
Schematization of the double-use action.

**Figure 2 materials-16-05436-f002:**
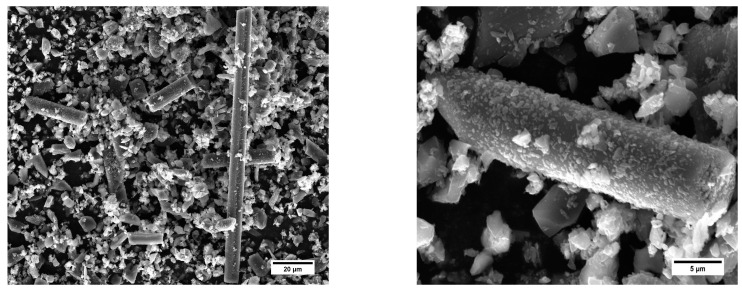
SEM micrographs of a sample of carbon microfibers (magnifications 3.75 k× (**left**) and 17.5 k× (**right**)).

**Figure 3 materials-16-05436-f003:**
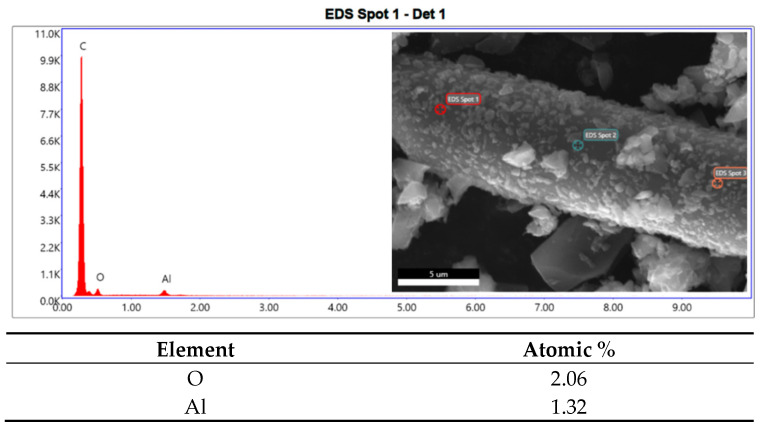
EDS analysis spectrum and atomic percentage of the elements.

**Figure 4 materials-16-05436-f004:**
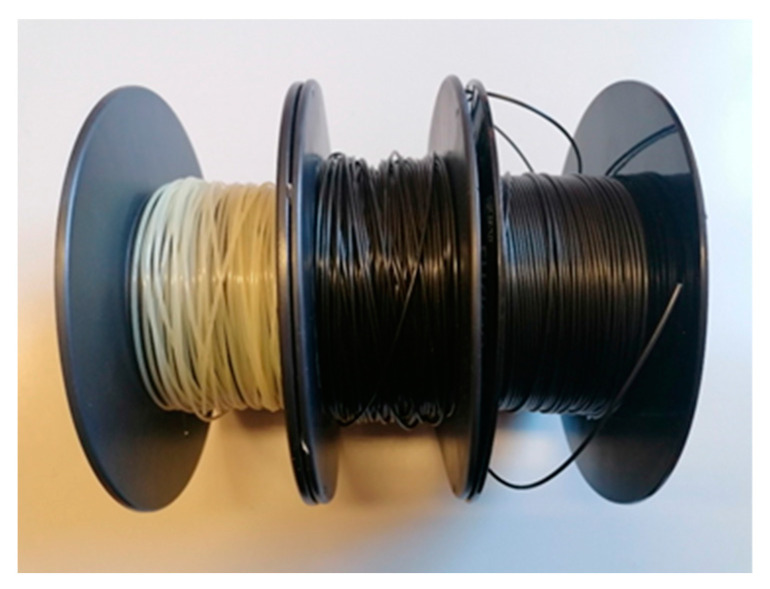
3D printing filaments obtained from extrusion processing.

**Figure 5 materials-16-05436-f005:**
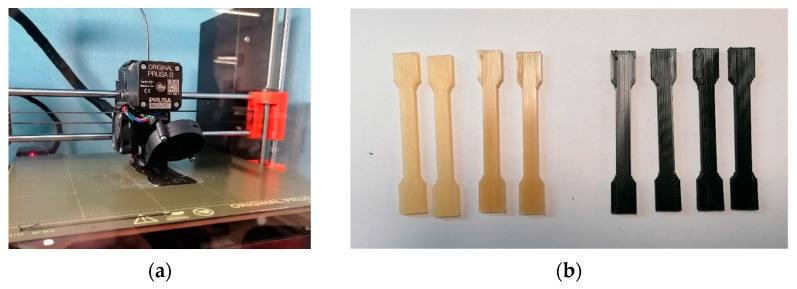
(**a**) PRUSA i3 3D printer and (**b**) 3D-printed dumbbell-shaped samples.

**Figure 6 materials-16-05436-f006:**
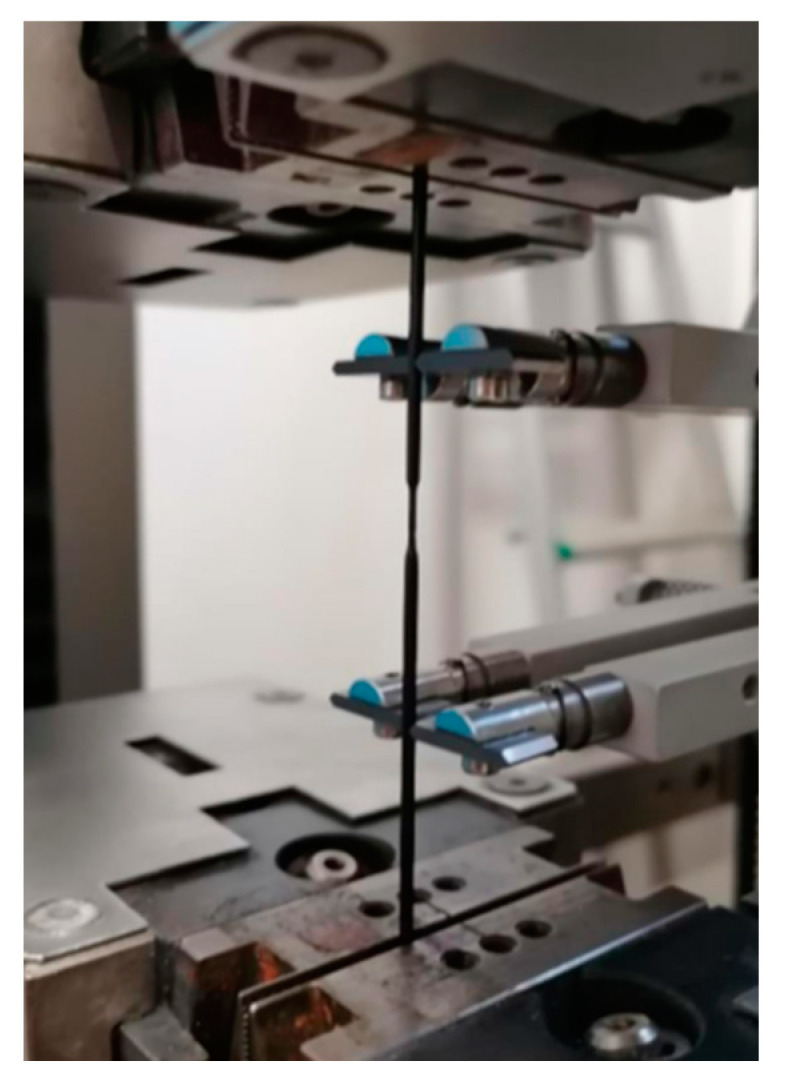
Tensile test of a filament.

**Figure 7 materials-16-05436-f007:**
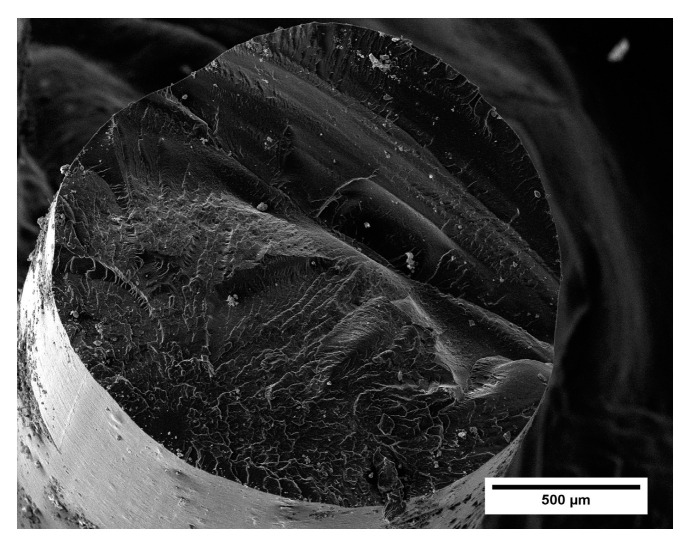
SEM micrograph of the neat PA6,6 filament (magnification 250×).

**Figure 8 materials-16-05436-f008:**
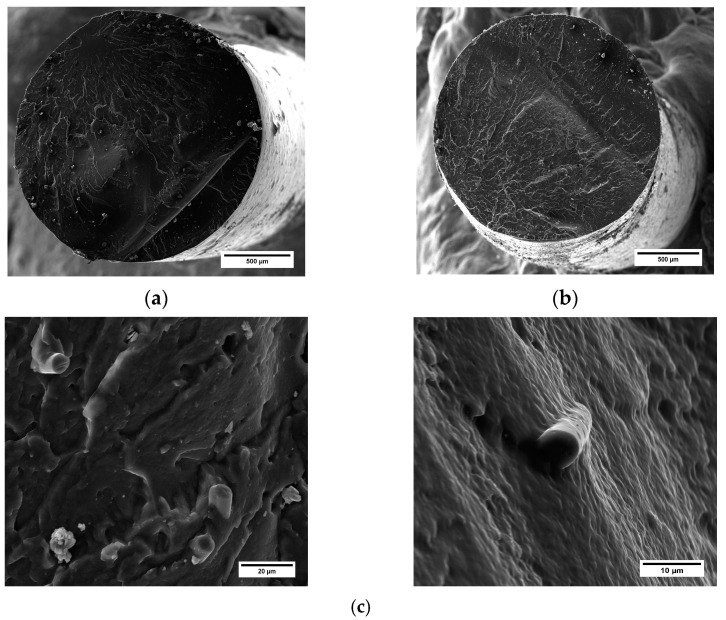
SEM micrograph of (**a**) PA6,6 + 5 wt.% rCF filament (magnification 250×) and (**b**) PA6,6 + 10 wt.% rCF filament (magnification 250×). (**c**) Details of rCFs embedded into the matrices (magnifications 5000× (left) and 8750× (right)).

**Figure 9 materials-16-05436-f009:**
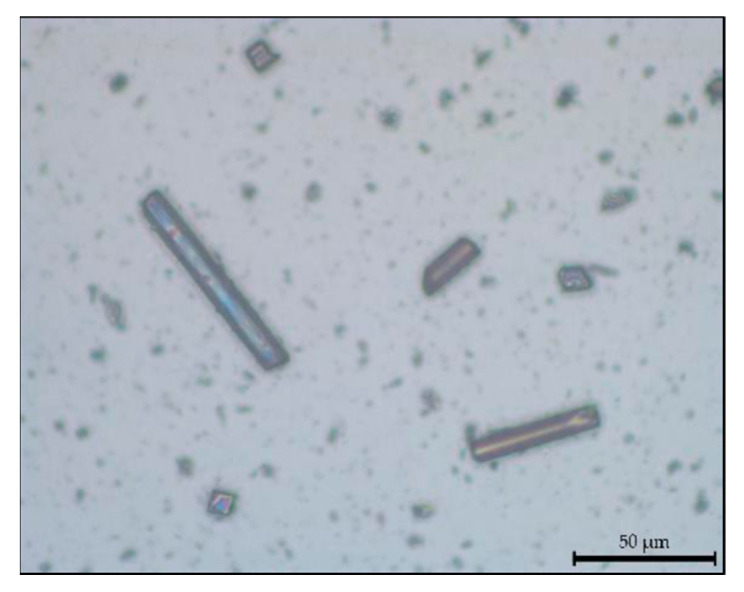
Carbon microfibers under the optical microscope (magnification 50×).

**Figure 10 materials-16-05436-f010:**
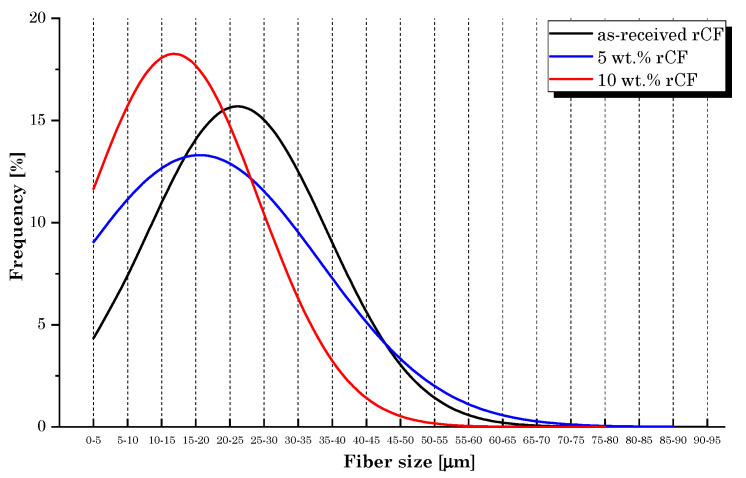
Fiber size distribution.

**Figure 11 materials-16-05436-f011:**
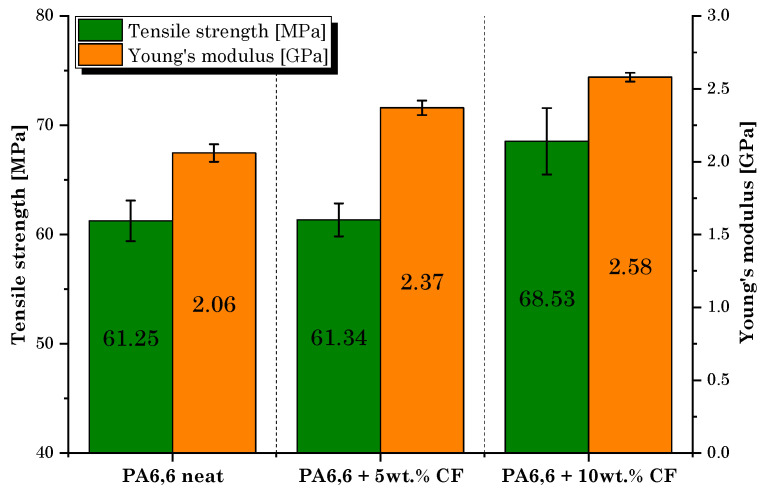
Tensile test results: tensile strength and Young’s modulus values of each manufactured filament.

**Figure 12 materials-16-05436-f012:**
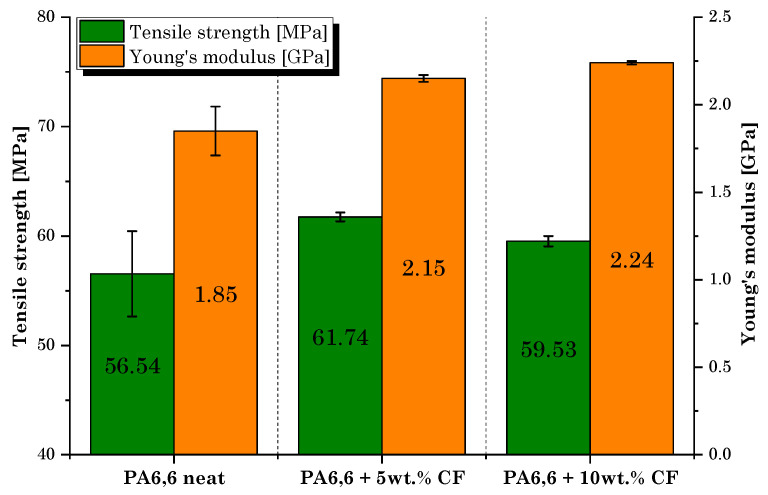
Tensile test results: tensile strength and Young’s modulus values of 3D-printed dumbbell-shaped samples.

**Figure 13 materials-16-05436-f013:**
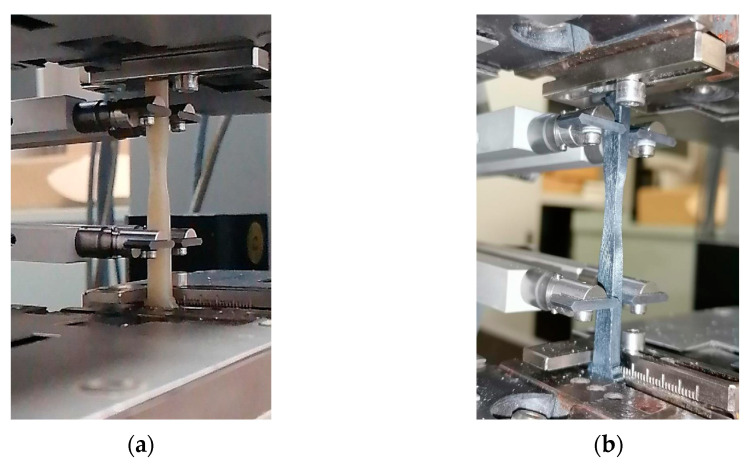
3D-printed composite specimens during tensile tests: (**a**) PA6,6 neat and (**b**) PA6,6 composite.

**Figure 14 materials-16-05436-f014:**
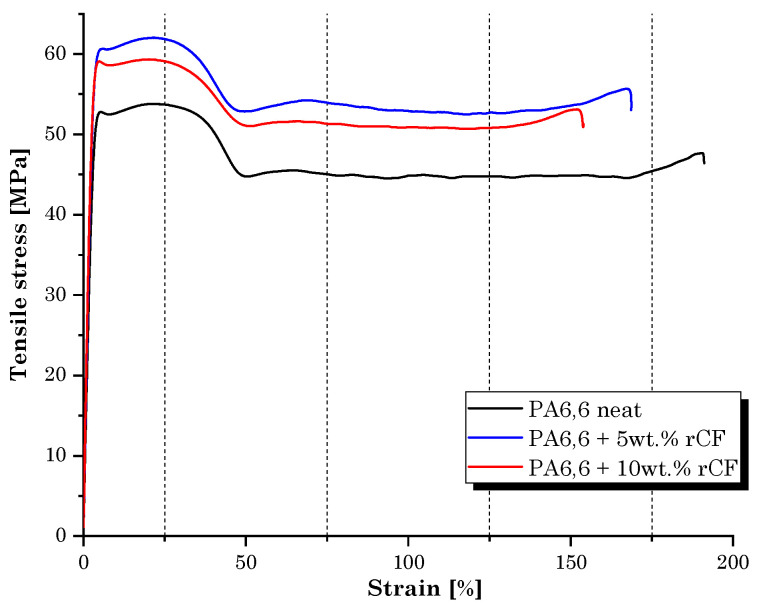
Stress–strain behavior of 3D-printed dumbbell-shaped samples.

**Figure 15 materials-16-05436-f015:**
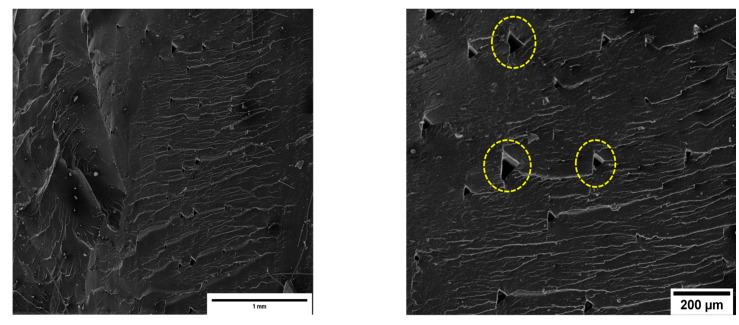
SEM micrographs of the neat PA6,6 specimen. The dotted circles highlight inter-filament voids in the matrix (magnifications 200× (**left**) and 500× (**right**)).

**Figure 16 materials-16-05436-f016:**
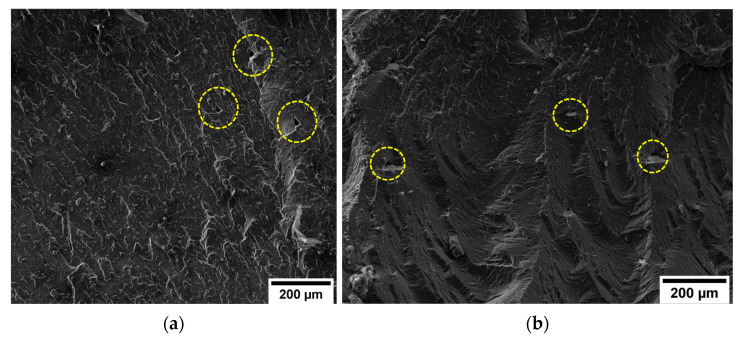
SEM micrographs (magnification 500×) of (**a**) PA6,6 + 5 wt.% rCF and (**b**) PA6,6 + 10 wt.% rCF specimens. The dotted circles highlight inter-filament voids in the matrix.

**Figure 17 materials-16-05436-f017:**
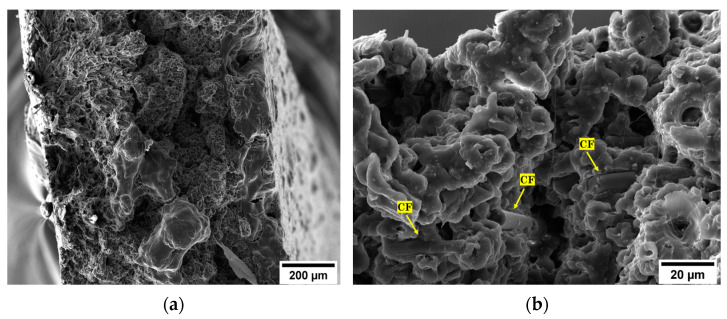
SEM micrographs of the fracture surface of the PA6,6 + 5 wt.% rCF specimen: (**a**) general view of the surface (magnification 500×) and (**b**) distribution of rCFs in the matrix (magnification 5 k×).

**Table 1 materials-16-05436-t001:** Physical and mechanical properties of BASF Ultramid^®^ 1000-11 NF2001 PA6,6 from the technical data sheet.

Properties	Values	Comments
Density	1.14 g/cc	ISO 1183-1
Water absorption	8.5%	ISO 62
Moisture absorption at equilibrium	2.5%	23 °C/50% R.H.; ISO 62
Tensile strength, yield	83.0 MPa	50 mm/min; ISO 527-1
Elongation at break	25%	50 mm/min, normal strain; ISO 527-1
Elongation at yield	5.0%	50 mm/min; ISO 527-1
Tensile modulus	3.00 GPa	1 mm/min; ISO 527-1
Flexural strength	117 GPa	ASTM D790 test
Flexural modulus	2.90 GPa	ASTM D790 test
Melting point	257 °C	10 K/min ASTM D3418 test

**Table 2 materials-16-05436-t002:** Temperature profiles used during the extrusion of PA6,6 filaments (neat and composites).

	PA6,6 Neat Filament	PA6,6 + rCF
Zone 1 temperature, °C	255	260
Zone 2 temperature, °C	255	260
Zone 3 temperature, °C	260	265
Zone 4 temperature, °C	260	265
Zone 5 temperature, °C	260	265
Zone 6 temperature, °C	255	260
Zone 7 temperature, °C	250	255
Die temperature, °C	235	240
Screw speed, rpm	150	150

**Table 3 materials-16-05436-t003:** Standard deviation of fiber dispersion in the specimens.

Sample	Mean ± Std. Dev.
PA6,6 neat	1.129 ± 0.026 g/cm^3^
PA6,6 + 5 wt.% rCF	1.148 ± 0.019 g/cm^3^
PA6,6 + 10 wt.% rCF	1.164 ± 0.014 g/cm^3^

**Table 4 materials-16-05436-t004:** Density values of CFs, neat PA6,6, and PA6,6 composites and estimated theoretical values of the percentage of reinforcement in composite filaments.

Material	Method	Value
rCFs	Helium pycnometer	1.917 g/cm^3^
PA6,6 neat filament	Buoyancy balance	1.129 g/cm^3^
PA6,6 + 5 wt.% rCF filament	Buoyancy balance	1.148 g/cm^3^
PA6,6 + 10 wt.% rCF filament	Buoyancy balance	1.164 g/cm^3^
Percentage of rCF reinforcement		
PA6,6 + 5 wt.% rCF filament	Rule of mixtures	2.41% *v*/*v*
PA6,6 + 10 wt.% rCF filament		4.44% *v*/*v*

**Table 5 materials-16-05436-t005:** Effective percentage of reinforcement in composite filaments obtained with an acid attack.

Material	Value
PA6,6 + 5 wt.% rCF filament	3.40% *v*/*v*
PA6,6 + 10 wt.% rCF filament	6.79% *v*/*v*

## Data Availability

Not applicable.

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
