# Peer review of "Carbon-Fiber-Recycling Strategies: A Secondary Waste Stream Used for PA6,6 Thermoplastic Composite Applications"

_materials, 2023, doi:10.3390/ma16155436_

Round 1

Reviewer 1 Report

Dear authors,

Overview and general recommendation:

Studying the possibility of carbon fibers recycling and application of microfibers as reinforcement in the thermoplastics is of particular interest in the development of composite materials. This topic is quite relevant and the current study is on a general interest to the readers of the journal; it is based on experimental results and compares them to the available from the literature. I find the paper to be much described and I am sure that authors performed it careful. On the other hand, I have seen that some points were unreasonable missing. Therefore, I recommend that a minor revision is warranted. I explain my remarks in more detail below.

1.      The objectives and goals of this study are not very clearly formulated in the introduction. Please add or reformulate more clearly the last few sentences (the rows 139-142) of the introduction.

2.      In the row 166 in signature of the Figure 1 the word "recovery" I recommend to change to "utilization".

3.      In row 246-247 the purpose of the fibers heating remained unclear. And “five filaments per formulation” is fuzzily.

4.      May be, the idea to put a scale or to add in signature of the figure 5 what is the diameter of the printed filaments have a meaning.

5.      Row 236 correct the paragraph indentation

6.      In the row 227-229, please, give a reference about principles of the measuring of density and buoyancy, method principles, accuracy.

7.      In Figure 4 and 5 the scale (ordinate) should starts at least from 40 MPa.

8.      Please check the English carefully.

Author Response

Cover letter for reviewer (Materials-2522777)

Reviewer 1

The authors would like to thank the reviewer for his valuable comments and suggestions on this manuscript. Below are our replies to comments.

  1. The objectives and goals of this study are not very clearly formulated in the introduction. Please add or reformulate more clearly the last few sentences (the rows 139-142) of the introduction.

The comment has been implemented as requested. The formulation of the objectives and goals of the study has been reformulated more clearly (see lines 132 to 151).

  1. In the row 166 in signature of the Figure 1 the word "recovery" I recommend to change to "utilization".

The comment has been implemented as requested (see line 176).

  1. In row 246-247 the purpose of the fibers heating remained unclear. And “five filaments per formulation” is fuzzily.

The composite specimens were treated in oven before testing to avoid the problems related to the moisture absorption of the PA6,6 matrix. The sentences were changed to make clearer the procedure description. The oven-drying was mentioned in other sections of the manuscript where need. The length of filament specimens for tensile test was also specified (see lines 234, 284, 307).

  1. May be, the idea to put a scale or to add in signature of the figure 5 what is the diameter of the printed filaments have a meaning.

According to the reviewer's comment, I believe that in this figure it is not necessary to insert a scale or directly the dimension of the diameter since it is already indicated in the lines preceding the figure and, moreover, it would not be indicative given the type of picture.

  1. Row 236 correct the paragraph indentation

The comment has been implemented as requested.

  1. In the row 227-229, please, give a reference about principles of the measuring of density and buoyancy, method principles, accuracy.

The comment has been implemented as requested in the row 251-266. More details and supporting references ([21] and [22]) have been added to the section to provide measuring principles and accuracy of the methods.

  1. In Figure 4 and 5 the scale (ordinate) should start at least from 40 MPa.

The plots in Figures 11 and 12 were edited in accordance with the reviewer’s comment.

  1. Please check the English carefully.

We have rechecked and reworded some parts with more accurate English, thanks for the suggestion.

Reviewer 2 Report

Reviewer Comments: Dear Respected Author, many thanks for sharing your Manuscript ID: materials-2522777, entitled "Carbon fiber recycling strategies: a second stream waste used  for thermoplastic composite applications". In my opinion, this manuscript can be accepted subject to the satisfactory round of manuscript.

1-    The problem statement is not good and novel. See In this article the problems related to the size of the fibers used and the presence of a preponderant dusty fraction will be addressed. Furthermore, the possible solutions to this  problem will be considered and the improvements in terms of mechanical properties will be evaluated.”

2-     "Introduction section is written superficially and there is no story between the paragraphs. There are so many paragraphs in the introduction section. The reviewer suggests merging some of them according to the main idea of each paragraph.

3-    Polish your introduction section. (please see annotated file)

4-    Redraw Figure 1 again, currently text and arrow lines overlap with each other

5-    Please highlight and annotate each SEM figure,  what are the readers supposed to notice in SEM images? Add more discussions in the text related to these images. Please merge some of the SEM figures as well.

6-    This manuscript has too many figures and some of them are reductant. For example, Remove  Figure 4, Figure 6, Figure 7, and Figure 12

7-    The reviewer suggests please reduce the aspect ratio of all figures. There is no need to use large images since your work is focused on recycling so why not give the proper impression through your work presentation? The current large image style is ironically occupied a large space and wastage of paper which is against the recycling theme. See Figure 14, figure 16 and Figure 15.

8-    The first letter of each figure and table must be capitalised in the text as well as in the caption. i.e, “Figure” and “Table”

9-    How many samples for each test were used?

1-   Validation of experimental results?   Numerical work?

1-   Please check for typos

1-   No future research scopes have been provided What areas need to explore? What kind of studies are required in future? How your study can be beneficial for future studies?

Best wishes,

Author Response

Cover letter for reviewer (Materials-2522777)

Reviewer 2

The authors would like to thank the reviewer for his valuable comments and suggestions on this manuscript. Below are our replies to comments.

  1. The problem statement is not good and novel. See” In this article the problems related to the size of the fibers used and the presence of a preponderant dusty fraction will be addressed. Furthermore, the possible solutions to this problem will be considered and the improvements in terms of mechanical properties will be evaluated.”

The text has been expanded and rewritten to highlight more clearly the problem statement and the novelty of the work (see lines 132 to 151).

  1. "Introduction section is written superficially and there is no story between the paragraphs. There are so many paragraphs in the introduction section. The reviewer suggests merging some of them according to the main idea of each paragraph.

In accordance with the reviewer’s comment, we carried out a merge between the paragraphs in the introduction section in order to create a common thread between the topics covered.

  1. Polish your introduction section. (Please see annotated file)

A supporting reference ([1]) to the sentence has been added as suggested by the reviewer in the annotated file (see line 35).

  1. Redraw Figure 1 again, currently text and arrow lines overlap with each other

The comment has been implemented as requested: the Figure 1 has been fixed.

  1. Please highlight and annotate each SEM figure, what are the readers supposed to notice in SEM images? Add more discussions in the text related to these images. Please merge some of the SEM figures as well.

We appreciate the reviewer's comment but in our opinion everything that can be deduced from the SEM images (filaments and 3D printed specimens) has been redacted and discussed in the original version of the manuscript. To broaden the discussion, a few additional remarks on filaments have been added (see lines 326 to 328 and 334). Furthermore, the SEMs of Figure 8 have been merged.

  1. This manuscript has too many figures and some of them are reductant. For example, Remove Figure 4, Figure 6, Figure 7, and Figure 12.

Figures 4 and 7 have been removed as requested, while we have left the other two. As for Figure 6, in our opinion is important to leave it in the article because it shows the correct deformation of the filament, proving the goodness of the test and the quality of the filament produced: necking occurs in the center, not near the grasps. Figure 9 is important because it shows and gives insight into how much the milling process reduces the initial fiber size, and this can lead us to upstream milling optimization to obtain less pulverulent fraction and longer fibers.

  1. The reviewer suggests please reduce the aspect ratio of all figures. There is no need to use large images since your work is focused on recycling so why not give the proper impression through your work presentation? The current large image style is ironically occupied a large space and wastage of paper which is against the recycling theme. See Figure 14, figure 16 and Figure 15.

The comment has been implemented as requested.

  1. The first letter of each figure and table must be capitalised in the text as well as in the caption. i.e, “Figure” and “Table”

The comment has been implemented. “Figure” and “Table” were capitalized.

  1. How many samples for each test were used?

The number of samples tested where missing in the manuscript was specified (see lines 266, 279, 287, 298, and 309).

10   Validation of experimental results?   Numerical work?

This work has not foreseen numerical analysis but has focused on the evaluation of the feasibility of incorporating a carbon waste filler to produce composite materials for advanced manufacturing processes. The advantages and criticalities of using this type of filler have been just “experimentally” explored. It has been mentioned in future developments to also perform thermal analysis of the composites to have a complete understanding of the material and process parameters. Implementing thermal analysis could involve DOE numerical analysis which however defer to future works.

11   Please check for typos

Done

12   No future research scopes have been provided What areas need to explore? What kind of studies are required in future? How your study can be beneficial for future studies?

The comment has been implemented by adding some details about future studies and applications area (see lines 662 to 674).

Best wishes,

Reviewer 3 Report

·         1. Introduction: It is necessary to highlight the aim of the manuscript more clearly. Also, in the Introduction, give a more comprehensive description of the topics and conducted research directly related to the topic of the paper. Include PA6.6 in the title of the paper and key words.

·        2.  Materials and Methods: Table 1. Standards need to be defined correctly (e.g. ISO 1183: -1, -2 or -3, ISO 527?, ASTM?). CFs production needs to be explained in more detail. Standards used for characterization of microfiber need to be defined. Description of the produced/extruded filaments (2.2.1) and 3D printing (2.2.7) must be specified in chapter 2.1 Materials. It is necessary to show the machine used for 3D printing. Pg. 6. How is the diameter of composite filaments determined? 2.2 Methods: Chapters 2.2.5 and 2.2.8 it is necessary to connect, as well as the chapters related to SEM analysis.  For SEM and optical microscopy magnification level is missing. For standardized test methods, it is necessary to correctly indicate the designation of the standard, describe the implementation of the test and the presentation of the results. For fibre-feinforced plastic composites, usually used tensile test is according to the ISO 527-4.

·       3. Results and discussion: Pg. 10 Describe the measurement of filament diameter and indicate the number of measurements taken. In figures 10 and 11 a. indicate the enlarged details shown in figures b. Indicate the magnification level on all microscopic images presented. 3.1.2 The introductory text/method description should be listed under the Methods. In Figures 14 and 15 add numerical values. Write that the average values were obtained by testing on (?) samples.

·         4. It is not usual to include values in formulas and display the result calculation. It is enough to give the formulas and calculated values.

·         5. Conclusion: Please correct the chapter number in 5. It is necessary to specify the possible application of the obtained materials.

Author Response

Cover letter for reviewer (Materials-2522777)

Reviewer 3

The authors would like to thank the reviewer for his valuable comments and suggestions on this manuscript. Below are our replies to comments.

  1. Introduction: It is necessary to highlight the aim of the manuscript more clearly. Also, in the Introduction, give a more comprehensive description of the topics and conducted research directly related to the topic of the paper. Include PA6.6 in the title of the paper and key words.

The aim of the manuscript and the conducted research within the topic of the paper have been highlighted more clearly and in more comprehensive way (see lines 132 to 151). PA6,6 was included in the title and keywords.

  1. Materials and Methods: Table 1. Standards need to be defined correctly (e.g. ISO 1183: -1, -2 or -3, ISO 527?, ASTM?). CFs production needs to be explained in more detail. Standards used for characterization of microfiber need to be defined. Description of the produced/extruded filaments (2.2.1) and 3D printing (2.2.7) must be specified in chapter 2.1 Materials. It is necessary to show the machine used for 3D printing. Pg. 6. How is the diameter of composite filaments determined? 2.2 Methods: Chapters 2.2.5 and 2.2.8 it is necessary to connect, as well as the chapters related to SEM analysis. For SEM and optical microscopy magnification level is missing. For standardized test methods, it is necessary to correctly indicate the designation of the standard, describe the implementation of the test and the presentation of the results. For fibre-reinforced plastic composites, usually used tensile test is according to the ISO 527-4.

Standard methods in Table 1 were defined correctly. The part concerning the production of rCFs has been rewritten to clarify the process of obtaining the filler in more detail (see lines 166 to 173). The ASTM standard for fiber density has been added (see line 179). For the other techniques implemented for fiber characterization (SEM, EDS, fiber size distribution) there are not specific standards. The description of the extruded filaments and 3D printing has been moved in Chapter 2.1 (Materials). The pic of the machine used for 3D printing has been added in Figure 5a. The filaments diameter was measured by digital caliper (the information has been added in line 220). We feel it is better and clearer to keep the discussion of SEM analyses (filaments and 3d printed samples) separate as in the original article. This is because the two types of investigation reveal information related to the mechanical performance of the filaments and 3D samples which were analysed separately. Magnification levels for SEM and optical microscopies have been added in the captions. The standard test method for 3D printed samples has been corrected in accordance with the reviewer’s comment (see line 305).

  1. Results and discussion: Pg. 10 Describe the measurement of filament diameter and indicate the number of measurements taken. In figures 10 and 11 a. indicate the enlarged details shown in figures b. Indicate the magnification level on all microscopic images presented. 3.1.2 The introductory text/method description should be listed under the Methods. In Figures 14 and 15 add numerical values. Write that the average values were obtained by testing on (?) samples.

The number of measurements and method for filament diameter (digital caliper) have been added by also inserting the average value and standard deviations (see line 330). The SEM details reported in the figures 10-11 b have been merged in a single figure (Figure 8c) showing the presence of fibers embedded into the matrix and their distribution. The magnification levels have been added in the captions. All test methods are now listed under Methods section. Numerical values as labels have been added in the figure reporting the results of tensile tests (see Figures 11 and 12). The proposed sentence and the number of samples tested in tensile have been specified in line 309.

  1. It is not usual to include values in formulas and display the result calculation. It is enough to give the formulas and calculated values.

The equations in section 4 have been modified to include only the result of the calculations

  1. Conclusion: Please correct the chapter number in 5. It is necessary to specify the possible application of the obtained materials.

The chapter number has been corrected (see line 638). The possible application of the obtained materials has been added and discussed in line 671.

Round 2

Reviewer 2 Report

Accepted

Acceptable

Reviewer 3 Report

 Accept in present form